# Anti-Aging and Lightening Effects of Au-Decorated Zeolite-Based Biocompatible Nanocomposites in Epidermal Delivery Systems

**DOI:** 10.3390/jfb14020066

**Published:** 2023-01-26

**Authors:** Seungyeon Lee, Geunjeong Lee, Giyoung Jeon, Hayeong Lee, Suhyeon Park, Youngju Sohn, Youngkum Park, Seongwoo Ryu

**Affiliations:** 1Department of Materials Science and Engineering, The University of Suwon, 17 Wauan-Gil, Bongdam-Eup, Hwaseong-si 18323, Gyeonggi-do, Republic of Korea; 2Gragem Co., Ltd. 21999, Room1009, 10F, Meet you all Tower Main 12, Geatbeol-ro, Yeonsu-gu, Incheon 21990, Republic of Korea

**Keywords:** Au-decorated zeolite, biocompatible nanocomposite, anti-aging, lightening

## Abstract

The main challenges in developing zeolites as cosmetic drug delivery systems are their cytotoxicities and the formation of drug-loading pore structures. In this study, Au-decorated zeolite nanocomposites were synthesized as an epidermal delivery system. Thus, 50 nm-sized Au nanoparticles were successfully deposited on zeolite 13X (super cage (α) and sodalite (β) cage structures) using the Turkevich method. Various cosmetic drugs, such as niacinamide, sulforaphane, and adenosine, were loaded under in vitro and in vivo observations. The Au-decorated zeolite nanocomposites exhibited effective cosmetic drug-loading efficiencies of 3.5 to 22.5 wt% under various conditions. For in vitro cytotoxic observations, B16F10 cells were treated with various cosmetic drugs. Niacinamide, sulforaphane, and adenosine-loaded Au-decorated zeolite nanocomposites exhibited clear cell viability of over 80%. Wrinkle improvement and a reduction in melanin content on the skin surface were observed in vivo. The adenosine delivery system exhibited an enhanced wrinkle improvement of 203% compared to 0.04 wt% of the pure adenosine system. The niacinamide- and sulforaphane-loaded Au-decorated zeolite nanocomposites decreased the skin surface melanin content by 123% and 222%, respectively, compared to 2 and 0.01 wt% of pure niacinamide and sulforaphane systems, respectively. As a result, Au-decorated zeolite nanocomposites show great potential as cosmetic drug epidermal delivery systems for both anti-aging and lightening effects.

## 1. Introduction

Recently, zeolites, which are biocompatible nanocomposites, have attracted attention due to their diverse structures, allowing for controlled and targeted drug delivery applications [1,2,3,4,5]. Zeolites, owing to their low cost, abundant natural availability, and mass production, have the advantage of commercial feasibility. Zeolites consist of various porous structures, including micropores, mesopores, and macropores. These pore structures allow for the delivery of different therapeutic agents to the targeted sites with controlled-release systems [6,7,8,9]. Silica-based mesoporous zeolites, particularly, have shown great potential for use in drug-release systems [10,11]. The ion-exchange ability and uniform structure of zeolites have direct benefits of absorbing and releasing organic or inorganic particles [9,12,13]. Due to the various sizes and hydrophilicity of drugs, the loading capacity and release rate are highly dependent on the pore sizes [14] and surface modification [15,16] of zeolites. However, there are several challenges in using zeolites as drug delivery systems. Because of the difference in hydrophilicity and pore sizes of the zeolite surface and drug molecules, the loading capacity could be limited, and the release rate could be too high. In addition, oxygenated functional groups of zeolites induce cytotoxicity and carcinogenic effects, resulting in the disruption of the cell structure as well as swelling in the mitochondria and squared cells [17,18,19].

An alternative to overcome these challenges is to introduce novel metal nanoparticles, such as Au, to zeolite surfaces. Au nanoparticles are ideal vehicles for targeted and selective drug delivery. Au nanoparticles have high biocompatibility, hydrophilicity, non-immunogenicity, and low toxicity [20,21,22]. Furthermore, cytotoxicity of Au nanoparticles could be controlled by their shape, size, and densities [23,24,25]. Various deposition methods, such as vapor-phase deposition and grafting, sol–gel, and ion-exchange methods, have been developed for deposition, precipitation, co-precipitation, and impregnation [26]. Au nanoparticles are also deposited by the Turkevich method, developed by Brust et al., which induces a reaction with AuCl_4_^−^ (using tetrachloroauric acid (HAuCl_4_)) and the reducing agent sodium borohydride in the presence of the desired ligand (thiol-terminated long-chain alkane) [27,28]. Depending on the method used, Au nanoparticles of up to 20 nm in diameter can be deposited. The loading efficiency of Au nanoparticles strictly depends on the form of zeolite and surface modification via ion exchangeability [29]. For instance, the Au loading efficiency is usually higher for the ammonium form than that of the hydrogen form of zeolite [30].

In this study, a biocompatible nanocomposite, Au zeolite, was developed for the delivery of various cosmetic drugs. Zeolite 13X was first calcinated to free the super cage (α) and sodalite (β) cage structures (Figure 1). The Au nanoparticles were subsequently decorated onto the zeolite nanocomposites by the Turkevich method to form Au zeolite. Finally, in vivo and in vitro observations were performed for three different molecules: niacinamide, sulforaphane, and adenosine. Niacinamide is a practical lightening substance recognized by the Ministry of Food and Drug Safety, which is known to cause skin-lightening by reducing the melanosome transfer from melanocytes to keratinocytes [31,32,33]. As well as hyaluronic-acid-based composites [34], sulforaphane is also an effective lightening agent that reduces melanin production and tyrosinase activity as an anti-inflammatory, antioxidant, and anti-cancer substance [35,36,37]. Adenosine is a wrinkle-improving agent supported by the Ministry of Food and Drug Safety and is known to enhance wound healing by binding to the A2A receptor and increasing collagen in the dermis [34,38].

## 2. Materials and Methods

### 2.1. Materials

Zeolite 13X powder and tetrachloroauric acid, used to decorate the zeolite 13X, were purchased from Sigma-Aldrich (MO, USA). Oleic acid (Sigma-Aldrich, MO, USA) was used to coat the zeolite after drug loading to reduce unnecessary release [39]. An alumina filter (pore size 0.2 μm, Cytiva, Seoul, Republic of Korea) was used for the filtering processes.

For in vitro cell culture and experimentation, B16F10 melanoma cells (Korea Cell Line Bank, Seoul, Republic of Korea), Human Dermal Fibroblast cells (HDF cells; Lonza, Switzerland), Dulbecco’s Modified Eagle Medium (DMEM; Capricorn Scientific, Ebsdorfergrund, Germany), Fetal Bovine Serum (FBS; Gibco Thermo Fisher Scientific, Waltham, MA, USA), Penicillin/Streptomycin (Gibco Thermo Fisher Scientific, Waltham, MA, USA), 3-(4,5-dimethylthiazol-2-yl)-2,5-diphenyltetrazolium bromide (Merck KGaA, Darmstadt, Germany), Alpha Melanocyte Stimulating Hormone (α-MSH, Merck KGaA, Darmstadt, Germany), 3-isobutyl-1-methylxanthine (IBMX; Merck KGaA, Darmstadt, Germany), NaNO_3_ (Sigma-Aldrich, MO, USA), NaOH (Duchefa Biochemie, Haarlem, The Netherlands), dimethyl sulphoxide (DMSO, Duchefa Biochemie, Haarlem, The Netherlands), Niacinamide (Merck KGaA, Darmstadt, Germany), Adenosine (Merck KGaA, Darmstadt, Germany), Sulforaphane (TCI, Tokyo, Japan), Trizol (Invitrogen Thermo Fisher Scientific, Waltham, MA, USA), Chloroform (Merck KGaA, Darmstadt, Germany), Isopropanol (Merck KGaA, Darmstadt, Germany), Ethanol (Merck KGaA, Darmstadt, Germany), and 3-(4,5-dimethylthiazol-2-yl)-2,5-diphenyltetrazolium bromide (MTT; Quanti-Max TM, Biomax, Seoul, Republic of Korea) were used.

### 2.2. Synthesis and Fabrication

#### 2.2.1. Preparation of Calcinated Zeolite

Zeolite 13X powder sized 3–5 μm, 10 Å pore size, and a bulk density greater than 0.61 g/mL was used. To increase the loading efficiency, the impurities in zeolite 13X were removed via calcination at a high temperature. The zeolite 13X powder was placed in a ceramic boat (85 mm × 50 mm × 20 mm), which was subsequently placed in an electric furnace (Hantech Co., Ltd. Gunpo-si, Gyeonggi-do, Republic of Korea). After loading into a vacuum environment, the temperature was increased (up to 450, 550, or 650 °C) at a rate of 2 °C/min and held for 6 h. After maintaining the process conditions for the required time interval, the temperature was decreased, and calcined zeolite was obtained.

#### 2.2.2. Synthesis of Au-Decorated Zeolite

The pH of the calcined zeolite 13X was controlled using a solution of 1 N NaNO_3_, 1 N NaOH, and distilled water. Thus, 1 N NaNO_3_ (1 L) was mixed with the zeolite, and the mixture was adjusted to pH 6 using 1 N NaOH. The pH-controlled zeolite was dispersed for 5 min using a sonicator (Ultrasonic Cleaner ABS, JAC-5020, Kodo, Hwaseong-si, Gyeonggi-do, Republic of Korea) and vacuum filtered using an alumina filter. Subsequently, the filtered powder was dried at 20 °C for 6 h. To decorate the zeolite with Au, 1.46 × 10^3^ M gold chloride hydrate (0.620 g) was mixed with distilled water (250 mL). The pH was adjusted to 6 using 1 N NaOH, and dried zeolite (2 g) was dispersed for 5 min using a sonicator. To decorate the zeolite surface and pores with Au, the mixture was placed on a hot plate (PC-420D, Corning, New York, NY, USA). The electromagnetic force was maintained at 200 rpm for 24 h. After stirring, the residual tetrachloroauric acid was washed thrice using distilled water, and the dispersion was vacuum filtered using an aluminum filter. After filtering, the resulting zeolite was dried for 24 h at room temperature.

#### 2.2.3. Fabrication of Zeolite Loaded with Active Materials

Each substance was loaded into zeolite pores. Before adding the materials, the entrapment efficiency of each material after filtering zeolite (1) was calculated using the following equation:
(1)$$\frac{(A+10)-B}{B}$$
where *A* represents the weight of the zeolite after filtering (g), 10 is the filtering loss (mg), and *B* is the weight of the zeolite materials before loading (g).

Table 1 lists the weight ratios obtained when materials were included. Efficiencies from 1 to 10 mg/mL were measured and compared by weight ratio.

Each active material was mixed with distilled water (100 mL) in appropriate ratios, and gold-decorated zeolite (2 g) was sonicated for 5 min and stirred at 200 rpm at 60 °C for 24 h. The dispersion was washed three times with distilled water to remove residual materials on the zeolite surface and vacuum filtered using an alumina filter. The retentate was dried at room temperature for 24 h.

#### 2.2.4. Coating of Zeolite with Organic Acid

The zeolite was coated with oleic acid to reduce unnecessary release. After mixing distilled water with 2–10 wt% oleic acid, 2 g of zeolite was added and stirred at room temperature for 15 min. The remaining oleic acid was separated by centrifugation (Mega 17R, Hanil Sci-Med, Daejeon, Republic of Korea) at 10,000 rpm for 10 min. After vacuum filtering with an alumina filter, the coated zeolite was dried at room temperature and stored at a temperature below the melting point of oleic acid (15 °C).

### 2.3. Characterization

#### 2.3.1. Microstructure and Loading Properties of Zeolite

Material characterization of the zeolite was performed using various equipment. Calcination of the zeolite was performed using an electric furnace (Hantech Co., Ltd. Gunpo-si, Gyeonggi-do, Republic of Korea). After fabricating Au-decorated zeolite and loading efficient materials, scanning electron microscopy (SEM, Apreo, FEI, Hillsboro, OR, USA) was performed to study the structure of zeolite 13X and at different calcination temperatures. Energy-dispersive X-ray spectroscopy (EDX, Apreo, FEI, Hillsboro, OR, USA) was used to confirm the uniformity of the zeolite surface, and transmission electron microscopy (TEM, CM200, Philips, Amsterdam, Netherlands) was used to confirm the loading efficiency by assessing the scattering peaks. The scale bar was set to 100 nm and Equation (2) was used to measure the surface area. The Na/Si ratio and uniformity of gold in the zeolite were measured using XPS (K-Alpha plus, Thermo Fisher Scientific, USA). The surface area and particle size were measured via physisorption (Brunauer–Emmett–Teller (BET), ASAP 2020, Micromeritics, GA, USA).
LamL = R, (R = 200 kV), (L = 30 cm)(2)

#### 2.3.2. Loading Efficiency Test

The absorbance of the materials in the zeolite was estimated using UV-vis-NIR spectrophotometry (Ultraviolet-Visible-Near-IR Spectroscopy, Lambda 750, Perkin Elmer, Waltham, MA, USA). The active materials were added to distilled water (100 mL) and stirred (comparison was performed after preparation by ratio). After the materials were dissolved, the absorbance of niacinamide, adenosine, and sulforaphane was measured at each concentration (0.2, 0.4, 0.6, 0.8, and 1.0 wt%). The peak was estimated for each material and a standard curve was generated. The loading efficiency of the coated zeolite onto the skin was measured using agarose gel electrophoresis. The coated zeolites were mixed into a permeated cream and immersed in agarose gel (3 g) for 5 min. The agarose gel was dissolved in deionized water (DIW) after removing the mixture from it, and the loading efficiency of the materials was measured.

#### 2.3.3. Cell Culture and Viability

B16F10 melanoma cells (Korea Cell Line Bank) and HDF (Lonza, Switzerland) were cultured in DMEM containing 10% FBS and 1% penicillin/streptomycin. The cells were cultured in a 5% CO_2_ incubator and maintained at 37 °C. MTT assay was used to determine the cytotoxicity of the samples. B16F10 melanoma cells were dispensed into a 96-well plate at a cell count of 1 × 10^4^ cells per well and treated with different concentrations of niacinamide, sulforaphane, adenosine, niacinamide + Au zeolite, sulforaphane + Au zeolite, and adenosine + Au zeolite, and incubated in a CO_2_ incubator at 37 °C for 24 h. Subsequently, a solution of 5 mg/mL MTT was added to each well, followed by incubation in an incubator (37 °C, 5% CO_2_) for 4 h. A microplate reader (Biotek Synergy-HT, USA) was used to measure absorbance at 540 nm. The experiment was conducted three times under the same conditions.

#### 2.3.4. Melanin Content Assay

B16F10 melanoma cells were dispensed into a 6-well plate at a cell count of 2 × 10^5^ cells per well and incubated for 24 h in DMEM containing 10% FBS. After 24 h of incubation, the culture medium was replaced with phenol-red-free DMEM. Subsequently, 100 μm IBMX and samples at each concentration were added and incubated for 72 h at 37 °C.

Following this, the supernatant from each well of the 6-well plate was transferred to a 96-well plate. The absorbance, measured at 490 nm using a microplate reader, was substituted into the standard melanin calibration curve to calculate the amount of extracellular melanin production.

After the supernatants were separated and detached, they were resuspended in PBS by centrifugation at 13,000 rpm for 5 min. The supernatants were removed, and 1 N NaOH and 10% DMSO were added to the cell pellet. The samples were boiled at 60 °C for 1 h, dissolved in melanin, and transferred to a 96-well plate. A microplate reader was used to measure the absorbance at 490 nm. The process was repeated three times to obtain an average value for calculating the amount of melanin produced in each well.

#### 2.3.5. Melanin Content and Wrinkle Improvement Assay In Vivo

The trials were conducted with more than 20 females over the age of 19 years, in a stable environment of constant temperature and humidity conditions (22 ± 2 °C and 50 ± 5%, respectively), in the absence of air current and direct sunlight. Each test subject stayed for 30 min under these conditions to ensure skin consistency.

Samples were prepared using GC-A-AG (Gragem cream (GC) with 0.04% adenosine (A) and 0.008% adenosine + Au zeolite (AG)), GC-B (Gragem cream basic) as the control, GC-N-NG (Gragem cream (GC) with 2.00% niacinamide (N) and 0.4% niacinamide + Au zeolite (NG)), and GC-S-SG (Gragem cream (GC) with 0.01% sulforaphane (S) and 0.002% sulforaphane + Au zeolite (SG)).

To measure the wrinkle depth, GC-A-AG was applied to the area under the left eye and GC-B was applied to the forehead; both creams were applied twice a day (morning and evening) during the final stage of the test subjects’ skincare routine. Average wrinkle depth was measured using Antera 3D (Miravex Ltd., Dublin, Ireland) before testing and after four weeks of application.

To measure melanin content, GC-B was used as a control. GC-N-NG and GC-S-SG were applied to the right cheek and GC-B was applied to the forehead; all three creams were applied twice a day (morning and evening) in the final stage of the test subjects’ skincare routine. Skin color intensity and total melanin concentration (surface + inner) on the skin were calculated using Mark-Vu (PSI Plus Co., Ltd., Republic of Korea) before testing and after four weeks of application.

#### 2.3.6. Statistical Analysis

All data were analyzed using one-way analysis of variance (ANOVA) for normally distributed values. Statistical significance was determined using one-way ANOVA followed by the Newman–Keuls multiple comparison test to analyze differences between the groups. Statistical analyses were performed using PRISM software (GraphPad Software, San Diego, CA, USA).

## 3. Results

### 3.1. Fabrication and Microstructure of Au-Decorated Zeolite

Au-decorated nanocomposites were fabricated using various cosmetic drugs, such as niacinamide, adenosine, and sulforaphane. The zeolite was calcined, as the calcination process can reduce the surface and pore impurities. After calcination, the zeolite 13X structure formed super-cage and beta-cage structures with hollow pore surfaces, allowing for the loading of various cosmetic drugs. The powder was maintained at temperatures up to 650 °C, and the particles were separated from the zeolite (Figure A1). To confirm the optimized temperature of the calcination process, the surface area of the zeolite at various temperatures (450, 550, and 650 °C) was observed. The surface areas were found to be 715.65 m^2^/g (450 °C), 738.80 m^2^/g (550 °C), and 661.01 m^2^/g (650 °C). Therefore, zeolite 13X was calcined at 550 °C due to the highest value of surface area at that temperature (Figure A2). Schematic 1 shows the structure of zeolite 13X before and after Au decoration using the Turkevich method. Au nanoparticles, with an average size of 50 ± 10 nm, were successfully nucleated and grown on the surface of zeolite 13X (Figure 1a,b). Figure 1c,d show the diffraction patterns of the zeolite and Au-decorated zeolite. The length of the dots in the diffraction pattern was measured, and the distance (Å) was subsequently calculated, allowing for the confirmation of the characteristic distance of gold materials, as shown in Figure 1d (2.35, 2.04, 1.44, and 1.23 Å). The distances obtained from the patterns confirm that the reflections of gold were at (111), (200), (220), and (311). Figure 1e,f show the X-ray photoelectron spectroscopy (XPS) survey of the zeolite and Au-decorated zeolite. The main atomic composition of zeolite 13X is Si, Al, and Na at 99.21 eV (15.93 at%), 73.31 eV (10.07 at%), and 1072.02 eV (8.72 at%), respectively. After Au decoration, the Au peak (83.5, 87.07 eV) was observed at 83.8 eV (9.03 at%). The reduced O contents at 531.75 eV (before Au decoration 53.22%, after Au decoration 51.94 %) indicates that Au nanoparticles were deposited on oxidative functional groups.

### 3.2. Pore-Size Distribution and Drug-Loading Release Efficiency of Au-Decorated Zeolite

To evaluate the efficiency of cosmetic drug loading and release of Au-decorated zeolite, the pore size and surface area were observed using BET. The BET surface area was measured to compare the zeolites before and after Au decoration. The surface area of zeolite was found to be 738.805 m^2^/g (Figure 2a) and 515.339 m^2^/g (Figure 2b) before and after Au decoration, respectively. During Au decoration, the pores of the zeolite were partially coated and filled with Au nanoparticles, reducing the surface area by 30.4%. The average pore size of the zeolite also differed after Au decoration. Before Au decoration, the average pore size was 54.796 Å (Figure 2c); however, after Au decoration, the average pore size increased to 60.591 Å (Figure 2d).

Three different cosmetic drugs were loaded onto the Au-decorated zeolite. Niacinamide, adenosine, and sulforaphane were loaded with oleic acid coating for entrapment. Among other fatty acids, such as capric acid, myristic acid, palmitic acid, and stearic acid, a small amount of oleic acid could reduce unnecessary release [40]. The loading efficiencies of niacinamide, adenosine, and sulforaphane were estimated using Equation 1 (Figure 2e). The loading efficiencies of niacinamide, adenosine, and sulforaphane were 10%, 4%, and 7%, respectively. Adenosine showed the highest loading efficiency, whereas niacinamide showed the lowest, mainly due to the differences in diffusion and adsorption on the zeolite surface. The delivery efficiency of each cosmetic drug was estimated via UV-vis absorption in an in vitro agarose gel. The reference for each material was first observed at different concentrations (Figure A3). Using a reference concentration, the delivery efficiency of each material on zeolite was estimated. Compared with the loaded cosmetic drugs, niacinamide, adenosine, and sulforaphane exhibited delivery efficiencies of 2.6%, 5.2%, and 2.7%, respectively.

### 3.3. Effect of Melanin Content In Vitro

The efficacy of the active niacinamide, sulforaphane, and adenosine materials and those collected in the biocompatible nanocomposite, Au zeolite, was compared alongside their effects on cell viability and melanin content. Data were expressed as the mean ± standard deviation of triplicate experiments (* *p* < 0.05, ** *p* < 0.01, and *** *p* < 0.001 vs. control group). First, the cell viability was assessed using an MTT assay. Subsequently, B16F10 cells (1 × 10⁴ cells/well) were cultured for 24 h and the medium was replaced with a serum-free medium, followed by treating the cells with different concentrations of the sample. Treatment with niacinamide at concentrations of 0.12, 0.60, 1.20, and 2.40 μg/mL yielded cell viability values of 95.0%, 92.7%, 87.8%, and 89.7%, respectively. Cell viabilities of 107.3%, 104%, 98%, and 88% were obtained after treating the Au zeolite with the same concentrations (Figure 3a). When sulforaphane and Au zeolite containing sulforaphane were treated at concentrations of 0.18, 0.80, 1.80, and 3.60 μg/mL, cell viabilities of 113.5%, 105.3%, 98.6%, and 82.9% were obtained for sulforaphane, whereas, for Au zeolite containing sulforaphane, they were 107.4%, 104.9%, 97.3%, and 87.5%, respectively (Figure 3b). Treating adenosine and Au zeolite containing adenosine at concentrations of 0.27, 1.35, 2.7, and 5.4 μg/mL yielded cell viabilities of 91.5%, 90.8%, 92.1%, and 84.9% for adenosine, and 114.2%, 100.1%, 91.0%, and 85.2% for Au zeolite containing adenosine, respectively (Figure 3c). Cell viabilities of 80% and higher, at all concentrations, were observed for each material.

B16F10 melanoma cells were dispensed into a 6-well plate at a cell count of 2 × 10^5^ cells per well and incubated for 24 h in DMEM containing 10% FBS. After 24 h of incubation, the culture medium was replaced with phenol-red-free DMEM. Cells treated with 100 μm IBMX(e) and 50 nm α-MSH(d) were used as a negative control to induce melanin formation at different sample concentrations.

The test resulted in a negative control group consisting of 50 nm α-MSH at 100% relative melanin content, compared to 23.8% for the control group. Sulforaphane, treated at a concentration of 0.18 and 1.8 μg/mL, caused melanin levels to decrease to 78.8% and 69.2%, respectively, in a concentration-dependent manner, demonstrating excellent lightening efficacy (Figure 3d). With 100 μm IBMX, the relative melanin content of the Au zeolite containing sulforaphane was 100%, whereas that of the control group was 17.2%. Au zeolite containing sulforaphane, treated with concentrations of 0.18, 0.80, 1.80, 3.60, and 1.8 μg/mL, resulted in 105.9%, 104.1%, 101%, and 97.7% relative melanin content, respectively (Figure 3e). This indicates a smaller drop in melanin; however, the concentration-dependent manner of Au zeolite containing sulforaphane was confirmed.

### 3.4. Effect of Melanin Content and Wrinkle Improvement In Vivo

The effect of the biocompatible nanocomposite, Au zeolite, as a drug delivery system was observed in terms of wrinkle depth and melanin content on the skin surface. The average wrinkle depth improvement rates in the test subjects for GC-B, GC-A, and GC-A-AG (***) were 5.97%, 8.46%, and 17.19%, respectively (Figure 4a) (*** *p* < 0.001 vs. control group). Wrinkle depth improvement relative to GC-B was 142% and 288% for GC-A and GC-A-AG, respectively (Figure 4b).

The average reduction in melanin on the skin surface of the test subjects for GC-B, GC-N, GC-N-NG, GC-S, and GC-S-SG was 0.92%, 1.36%, 1.67%, 0.81%, and 1.70%, respectively (Figure 4c). The means for GC-N and GC-N-NG, the reduction in melanin on the skin surface relative to GC-B was 148% and 181%, respectively (Figure 4d). The relative average melanin content on the skin surface for GC-S and GC-S-SG was 88% and 185%, respectively (Figure 4d).

The average reduction rates in melanin content on the skin of the subjects for GC-B, GC-N, GC-N-NG (*), GC-S, and GC-S-SG (**) were 20.30%, 26.10%, 28.81%, 26.00%, and 30.10%, respectively (Figure 4e) (* *p* < 0.05, ** *p* < 0.01 vs. control group). The means for the reduction in melanin on the skin surface relative to GC-B, GC-N, and GC-N-NG were 129% and 142%, respectively (Figure 4f). Furthermore, the relative average melanin content on the skin for GC-S and GC-S-SG was 128% and 148%, respectively (Figure 4f).

## 4. Discussion

The microstructure of Au-decorated zeolite was observed by TEM and XPS analysis (Figure 1). According to TEM images and diffraction patterns, 50 nm Au nanoparticles were successfully decorated on the free super-cage (α) and sodalite (β) cage structures of zeolite 13X via calcination and the Turkevich method. XPS analysis indicates that approximately 10% Au nanoparticles were successfully decorated onto the surface of zeolite 13X. Furthermore, the pore-size distribution shown in Figure 2 indicates that after Au decoration, Au nanoparticles only reduce the number of micropores less than 20 Å. Therefore, the effective drug-loading pores were not affected by Au decoration. As a result, the Au-decorated zeolite nanocomposites exhibited effective cosmetic drug-loading efficiencies of 3.5% to 22.5% under various conditions.

The Au-decorated zeolite nanocomposites exhibited safe cytotoxicity, owing to the novel metal decoration. Au-decorated zeolite 13X showed in vitro cell viabilities of 80% and higher, at all concentrations (Figure 3). Each material indicated that the biocompatible nanocomposite, Au zeolite, is a safe material, and the lightening efficacy of sulforaphane and Au zeolite material containing sulforaphane was evaluated at concentrations without cytotoxicity. Finally, the effect of melanin content and wrinkle improvement was observed as in vivo observation (Figure 4). Adenosine and niacinamide delivery systems were observed to have a lightening effect in vivo, whereas niacinamide and sulforaphane delivery systems were observed to have anti-aging effects. Improvement in wrinkles and reductions in melanin content on the skin surface were observed in vivo. The adenosine delivery system exhibited an enhanced wrinkle depth improvement of 203% compared to 0.04 wt% in the pure adenosine system. The niacinamide- and sulforaphane-loaded Au-decorated zeolite nanocomposites decreased the skin surface melanin content by 123% and 222%, respectively, compared to 2 and 0.01 wt% by pure niacinamide and sulforaphane systems, respectively. The Au-decorated zeolite nanocomposites in this study suggest an effective route for a safe epidermis drug delivery system, which opens the cosmetic potential for anti-aging and lightening applications.

## 5. Conclusions

In summary, a biocompatible nanocomposite, Au-decorated zeolite, showed effective cosmetic drug epidermal delivery systems for both anti-aging and lightening effects. By decorating Au nanoparticles on zeolite, we reduced cytotoxicity of zeolite without reducing available drug-loading pores. As a result, both cytotoxicity and drug delivery efficiency were dramatically enhanced for both in vitro and in vivo observation. Over 80% of cell viability was achieved as in vivo observation. Furthermore, 203% improved wrinkle depth and 222% decreased melamine contents were achieved in in vivo observation. The Au-decorated zeolite nanocomposites in this study suggest an effective route for a safe epidermis drug delivery system, which opens the cosmetic potential for anti-aging and lightening applications.

## Data Availability

The authors have supplementary data to share in the Appendix A.

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
