# Peer review of "Anti-Aging and Lightening Effects of Au-Decorated Zeolite-Based Biocompatible Nanocomposites in Epidermal Delivery Systems"

_jfb, 2023, doi:10.3390/jfb14020066_

Round 1
Reviewer 1 Report
Respected authors,
I have very much enjoyed your manuscript entitled "Anti-aging and whitening effects of Au-decorated zeolite-based biocompatible nanocomposites in epidermal delivery systems". I found your approach very interesting, especially considering the remarkable enhancement of desired functionality you aimed for and that you achieved!
First of all, I have two very small observations, which I think are related to typos:
- line 156 - in section 2.3.1 you mention "Mechanical property"; there is no mechanical analysis (of course, it would not be relevant to the study);
- line 254 - please, double check the calcination temperatures and the surface area values.
On the other hand, I think the discussion regarding cytotoxicity should be slightly enriched.
I am intrigued by the gold decoration step you carried out, in order to "reduced cytotoxicity of zeolite" - line 404 in the Conclusions section. Which is the foundation of this affirmation, is is solely based on literature reports on similar studies and known effects of Au nanoparticles decoration? I find this peculiar because if you actually studied bioactive principles + pristine zeolite systems, 2.3.3 Cell culture and viability does not really indicate it (since all samples you list seem to be Au decorated - lines 192-193).
Regardless the case, I think you should briefly discuss why Au nanoparticles improve the cytocompatibility of the zeolite substrate, especially in your kind of systems which contain approx. 9 at%. When also considering TEM, the surface of the decorated zeolites seems to be barely covered with Au nanoparticles (as it should, of course, to ensure access to the pore network). Which could possibly be the mechanism that ensures this positive effect?
In addition, I also think oleic acid coating should be discussed when approaching this compatibility issue, since it might impact to some extent cell-zeolite platform interactions by creating a smoother interface, less disruptive for cell membrane? And since we are on the topic of oleic acid, which was the rationale behind this choice? Could drug loading efficiency be influenced by another fatty acid (longer-/shorter- chained and or unsaturated)?
Independently on the issue of cytocompatibility, but still regarding the oleic acid:
- did the coating medium consisted only of water? Was the oleic acid homogeneously distribututed within the volume of water, without the need of some organic solvent?
- which should be the bonding site onto the zeolite substrate of the (I assume) carboxilic acid? Or what is the type of interaction that ensures the coating of the zeolite platform with oleic acid?
Author Response
We appreciate the comment by reviewer responses to all of the comments from the reviewer are below. In addition, our major changes were marked with red texts in the revised manuscript.
â—Ž Responses to Review
Reviewer 1
Comment 1: have very much enjoyed your manuscript entitled "Anti-aging and whitening effects of Au-decorated zeolite-based biocompatible nanocomposites in epidermal delivery systems". I found your approach very interesting, especially considering the remarkable enhancement of desired functionality you aimed for and that you achieved!
First of all, I have two very small observations, which I think are related to typos:
- line 156 - in section 2.3.1 you mention "Mechanical property"; there is no mechanical analysis (of course, it would not be relevant to the study);
We appreciate for your interest in our research works. As mechanical properties are not discussed in this study, we revised word ‘mechanical property’ to ‘microstructure’.
Our modification:
line 157
Microstructure
-line 254 - please, double check the calcination temperatures and the surface area values.
As reviewer suggested, we double checked calcination temperatures and the surface area values. We revised miss typed calcination temperature written in the manuscript.
Our modification:
line 254
650 °C
Comment 2: On the other hand, I think the discussion regarding cytotoxicity should be slightly enriched.
I am intrigued by the gold decoration step you carried out, in order to "reduced cytotoxicity of zeolite" - line 404 in the Conclusions section. Which is the foundation of this affirmation, is solely based on literature reports on similar studies and known effects of Au nanoparticles decoration? I find this peculiar because if you actually studied bioactive principles + pristine zeolite systems, 2.3.3 Cell culture and viability does not really indicate it (since all samples you list seem to be Au decorated - lines 192-193).
Regardless the case, I think you should briefly discuss why Au nanoparticles improve the cytocompatibility of the zeolite substrate, especially in your kind of systems which contain approx. 9 at%. When also considering TEM, the surface of the decorated zeolites seems to be barely covered with Au nanoparticles (as it should, of course, to ensure access to the pore network). Which could possibly be the mechanism that ensures this positive effect?
As reviewer suggested, we further discussed why Au nanoparticles improve the cytocompatibility of the zeolite substrate. The cytotoxicity is closely related to remained oxygenated functional groups of zeolite. Therefore, our strategy is to minimize remained oxygenated functional group of zeolite with Au nanoparticles. Cytotoxicity of Au nanoparticle is highly depends on their shape (J Mater Sci Mater Med. 2019; 30(2): 22.) and sizes (Journal of Photochemistry and Photobiology B: Biology 2020, 203, 111778). For example, sphere shaped Au nanoparticle exhibits minimum cytotoxicity. Once low density Au nanoparticles are applied to the numerous functional groups, higher proliferation rate of cells were observed (Coatings 2020, 10(9), 802). We assume that Au nanoparticle are directly applied to oxygenated functional group of zeolite to minimized cellular toxicity.
Our modification:
line 48
In addition, oxygenated functional groups of zeolites induce cytotoxicity and carcinogenic effects, resulting in the disruption of the cell structure as well as swelling in the mitochondria and squared cells [17, 18, 19].
line 54
Furthermore, cytotoxicity of Au nanoparticles could be controlled by their shape, size and density [23, 24, 25].
Ref.
[23] Steckiewicz, K. P.; Barcinska, E.; Malankowska, A.; Zauszkiewicz–Pawlak, A.; Nowaczyk, G.; Zaleska-Medynska, A.; Inkielewicz-Stepniak, I. Impact of gold nanoparticles shape on their cytotoxicity against human osteoblast and osteosarcoma in in vitro model. Evaluation of the safety of use and anti-cancer potential. Journal of Materials Science: Materials in Medicine, 2019, 30(2), 1-15.
[24] Chakraborty, A.; Das, A., Raha, S.; Barui, A. Size-dependent apoptotic activity of gold nanoparticles on osteosarcoma cells correlated with SERS signal. Journal of Photochemistry and Photobiology B: Biology, 2020, 203, 111778.
[25] Aysin, F.; Yilmaz, A.; Yilmaz, M. Metallic Nanoparticle-Decorated Polydopamine Thin Films and Their Cell Proliferation Characteristics. Coatings, 2020, 10(9), 802.
line 269
The reduced O contents at 531.75 eV (before Au decoration 53.22 %, after Au decoration 51.94 %) indicates that Au nanoparticles were deposited on oxidative functional groups.
Comment 3: In addition, I also think oleic acid coating should be discussed when approaching this compatibility issue, since it might impact to some extent cell-zeolite platform interactions by creating a smoother interface, less disruptive for cell membrane? And since we are on the topic of oleic acid, which was the rationale behind this choice? Could drug loading efficiency be influenced by another fatty acid (longer-/shorter- chained and or unsaturated)?
As reviewer suggested, fatty acid may influence the loading effect. However, we coated oleic acid after drug loading to reduce unnecessary release. Therefore, we assume that oleic acid did not directly affect drug loading. Also, to minimize the effect of fatty acid, we used oleic acid. Among other fatty acid such as capric acid, myristic acid, palmitic acid, and stearic acid, oleic acid can be coated in the smallest amount that minimizing the effects of fatty acids (Colloids and Surfaces A 606 (2020) 125371, Journal of Industrial and Engineering Chemistry, Volume 24, 25 April 2015, Pages 181-187). As reviewer suggested, we revised following discussion in the manuscript and we will discover effect of fatty acid in the future research works.
Our modification:
line 82
Oleic acid (Sigma Aldrich, MO, USA) was used to coat the zeolite after drug loading to reduce unnecessary release [39].
line 289
Among other fatty acids such as capric acid, myristic acid, palmitic acid, and stearic acid, small amount of oleic acid could reduce unnecessary release [40].
Ref.
[39] 39. Ong, H. T.; Suppiah, D. D.; Julkapli, N. M. Fatty acid coated iron oxide nanoparticle: effect on stability, particle size and magnetic properties. Colloids and Surfaces A: Physicochemical and Engineering Aspects, 2020, 606, 125371.
[40] 40. Dey, A.; Purkait, M. K. Effect of fatty acid chain length and concentration on the structural properties of the coated CoFe2O4 nanoparticles. Journal of Industrial and Engineering Chemistry, 2015, 24, 181-187.
Comment 4: Independently on the issue of cytocompatibility, but still regarding the oleic acid:
Did the coating medium consisted only of water?
Yes, we only used water as coating medium.
Comment 5: Was the oleic acid homogeneously distribututed within the volume of water, without the need of some organic solvent?
A small amount of oleic acid is enough for homogenous dispersion. Other previous researchers also reported uniform dispersion without any additional organic solvent nor surfactants (Colloids and Surfaces A 606 (2020) 125371’).
Comment 6: Which should be the bonding site onto the zeolite substrate of the (I assume) carboxilic acid? Or what is the type of interaction that ensures the coating of the zeolite platform with oleic acid?
Compare to a high Si/Al ratio zeolites, the zeolite 13x has hydrophilic properties that allows to attract polar groups of oleic acid.

Reviewer 2 Report
The authors undertook the Anti-aging and whitening effects of Au-decorated zeolite-based biocompatible nanocomposites in epidermal delivery systems.
Why authors choose expensive Au nanoparticles???
There are several polymeric composite that have antiageing effects like HA.
like https://doi.org/10.1016/j.jiec.2018.05.007. Please explain.
Author Response
We appreciate the comment by reviewer responses to all of the comments from the reviewer are below. In addition, our major changes were marked with red texts in the revised manuscript.
â—Ž Responses to Review
Reviewer 2
Comment 1: The authors undertook the Antiaging and whitening effects of Au-decorated zeolite-based biocompatible nanocomposites in epidermal delivery systems. Why authors choose expensive Au nanoparticles?
The Au is one of the well know noble metal that cytotoxicity of Au nanoparticle could be controlled by their shape (J Mater Sci Mater Med. 2019; 30(2): 22.) and sizes (Journal of Photochemistry and Photobiology B: Biology 2020, 203, 111778). Furthermore, by applying low density Au nanoparticles on numerous oxidative functional groups, we could induce higher proliferation rate of cells on the surface (Coatings 2020, 10(9), 802). As reviewer suggested, we added following discussion in the manuscript.
Our modification:
line 49
In addition, oxygenated functional groups of zeolites induce cytotoxicity and carcinogenic effects, resulting in the disruption of the cell structure as well as swelling in the mitochondria and squared cells [17, 18, 19].
line 54
Furthermore, cytotoxicity of Au nanoparticles could be controlled by their shape, size and density [23, 24, 25].
Ref.
[23] Steckiewicz, K. P.; Barcinska, E.; Malankowska, A.; Zauszkiewicz–Pawlak, A.; Nowaczyk, G.; Zaleska-Medynska, A.; Inkielewicz-Stepniak, I. Impact of gold nanoparticles shape on their cytotoxicity against human osteoblast and osteosarcoma in in vitro model. Evaluation of the safety of use and anti-cancer potential. Journal of Materials Science: Materials in Medicine, 2019, 30(2), 1-15.
[24] Chakraborty, A.; Das, A., Raha, S.; Barui, A. Size-dependent apoptotic activity of gold nanoparticles on osteosarcoma cells correlated with SERS signal. Journal of Photochemistry and Photobiology B: Biology, 2020, 203, 111778.
[25] Aysin, F.; Yilmaz, A.; Yilmaz, M. Metallic Nanoparticle-Decorated Polydopamine Thin Films and Their Cell Proliferation Characteristics. Coatings, 2020, 10(9), 802.
line 267
The reduced O contents at 531.75 eV (before Au decoration 53.22 %, after Au decoration 51.94 %) indicates that Au nanoparticles were deposited on oxidative functional groups.
Comment 2: There are several polymeric composite that have antiageing effects like HA.
like https://doi.org/10.1016/j.jiec.2018.05.007. Please explain.
As reviewer suggested, we included following reference to the introduction.
Our modification:
line 73
As well as hyaluronic acid based composites [34], sulforaphane is also an effective whit-ening agent that reduces melanin production and tyrosinase activity as an an-ti-inflammatory, antioxidant, and anti-cancer substance [35, 36, 37].
Ref.
[34] Zhang, J. N.; Chen, B. Z.; Ashfaq, M.; Zhang, X. Pe.; Guo, X. D. Development of a BDDE-crosslinked hyaluronic acid based microneedles patch as a dermal filler for anti-ageing treatment, Journal of Industrial and Engineering Chemist 2018, 65, 363–369.
